# Preliminary Findings of the Role of FAPi in Prostate Cancer Theranostics

**DOI:** 10.3390/diagnostics13061175

**Published:** 2023-03-19

**Authors:** Riccardo Laudicella, Alessandro Spataro, Ludovica Crocè, Giulia Giacoppo, Davide Romano, Valerio Davì, Maria Lopes, Maria Librando, Antonio Nicocia, Andrea Rappazzo, Greta Celesti, Flavia La Torre, Benedetta Pagano, Giuseppe Garraffa, Matteo Bauckneht, Irene A Burger, Fabio Minutoli, Sergio Baldari

**Affiliations:** 1Nuclear Medicine Unit, Department of Biomedical and Dental Sciences and Morpho-Functional Imaging, University of Messina, 98124 Messina, Italy; 2Nuclear Medicine Unit, S. Antonio Abate Hospital, 91016 Trapani, Italy; 3Nuclear Medicine, IRCCS Ospedale Policlinico San Martino, 16132 Genova, Italy; 4Department of Health Sciences (DISSAL), University of Genova, 16132 Genova, Italy; 5Department of Nuclear Medicine, Cantonal Hospital Baden, 5404 Baden, Switzerland; 6Department of Nuclear Medicine, University Hospital Zurich, University of Zurich, 8091 Zurich, Switzerland

**Keywords:** fibroblast activation protein, prostate, PET, PRRT, RLT

## Abstract

Prostate cancer (PCa) is the most frequently diagnosed cancer worldwide and the second most common cause of cancer-related deaths among men. Progress in molecular imaging has magnified its clinical management; however, an unmet clinical need involves the identification of new imaging biomarkers that complement the gold standard of prostate-specific membrane antigen (PSMA) positron emission tomography (PET) in cases of clinically significant PCa that do not express PSMA. Fibroblast activation protein (FAP) is a type II transmembrane serine overexpressed in many solid cancers that can be imaged through quinoline-based PET tracers derived from an FAP inhibitor (FAPi). Preliminary results of FAPi application in PCa (in PSMA-negative lesions, and in comparison with fluorodeoxyglucose—FDG) are now available in the literature. FAP-targeting ligands for PCa are not limited to detection, but could also include therapeutic applications. In this preliminary review, we provide an overview of the clinical applications of FAPi ligands in PCa, summarising the main results and highlighting contemporary strengths and limitations.

## 1. Introduction

Prostate cancer (PCa) is the most frequently diagnosed cancer worldwide and the second most common cause of cancer-related deaths among men [1]. Prostate-specific antigen (PSA) is the most useful biochemical parameter in PCa clinical practice [2] with a more limited value at the advanced stage due to increased PCa heterogeneity and de-differentiated lesions, also considering that optimal intervals for PSA follow-up are unknown [3]. Fortunately, the progress of next-generation imaging in the past few years has revolutionised PCa management. For example, using tracers that target the prostate-specific membrane antigen (PSMA) in positron emission tomography (PET) has become a well-established technique for the assessment of biochemical recurrence (BCR) [4], as well as further applications for biopsy guidance [5], staging [6] and the evaluation of disease progression [7]. However, the diagnostic accuracy of PSMA PET is challenged by findings that claim that up to 15% of clinically relevant PCa cases do not express this protein [8], leading to false negative results. Moreover, further cancer de-differentiation in the metastatic castration-resistant (mCR) late stages of the disease increases this percentage [9,10]. Therefore, identifying alternative imaging biomarkers that complement PSMA PET in such clinical circumstances represents an unmet clinical need.

In the last few years, the tumour microenvironment (TME) has gained attention due to its role in tumourigenesis, neo-angiogenesis and cancer progression. The TME consists of immune cells, extracellular matrix, vasculature, and cancer-associated fibroblasts (CAFs), which are the primary stromal cells within the TME. They can be identified based on the expression of several markers [11], including the fibroblast activation protein (FAP), which can be targeted by quinoline-based PET tracers derived from an FAP inhibitor (FAPi) [12]. FAP is a type II transmembrane serine protease that is nearly absent in healthy tissues, benign tumours, and normal fibroblasts (except in cases of chronic inflammatory conditions). However, in over 90% of the most common human epithelial tumours, it is overexpressed in the stroma and is histologically characterised by desmoplastic reactions [12]. FAP overexpression increases the risk of tumour invasion, lymph node metastasis and the decreased overall survival (OS) [12,13,14,15] of many solid cancers, including mCRPCa [16], especially if the cancer exhibits neuroendocrine (NE) differentiation [17]. It thus represents a potential target for theranostics applications for aggressive (and often, low-PSMA-expressing) PCa [13,18,19,20,21,22,23,24,25]. In this systematic review, we provide an overview of clinical applications using FAPi ligands in the PCa theranostics scenario by summarising the main preliminary results.

## 2. Materials and Methods

A literature search was performed by six independent authors (R.L., V.D., M.L., A.N., A.R. and G.C.) using the PubMed, Scopus, Google Scholar, Cochrane and EMBASE databases regarding the role of FAPi PET in PCa. The following terms were used: “FAP” or “fibroblast activation protein” or “FAPi” and “prostate cancer” or “prostate” and “PET” or “positron emission tomography” or “RLT” or “PRRT”. Papers (including original articles, clinical studies, and case reports of humans) that were written in the English language and had been published before September 2022 were included in the review. Editorials and reviews were excluded. Duplicate papers were removed. All papers were screened by title and abstract content. Full texts of papers that were inherent to the endpoint of the present review were retrieved to verify their relevance. Moreover, to enrich the collected data, all the references in the selected papers were also checked. The following data were extracted from the selected papers: name of the first author, year of publication, type of FAPi tracer used, sample size, number of PCa, the scenario, International Society of Urological Pathology (ISUP), prostate-specific antigen (PSA) values (when these were available) and the main findings.

## 3. Results

The literature search returned 65 studies. Five duplicate papers were eliminated. After screening the titles and abstracts, 36 full-text papers were retrieved. Of these, 21 articles did not fulfil the selection criteria and were excluded. Figure 1 details the selection process workflow according to the PRISMA guidelines [26].

The 15 remaining articles were published by researchers from Europe and Asia whose studies were conducted on 1–6 patients with PCa who underwent FAPi PET or FAPi radioligand therapy (RLT). Tables 1–4 describe the studies’ main characteristics and results according to the considered PCa scenarios.

## 4. Discussion

### 4.1. FAPi Immunohistochemistry (IHC) and Biodistribution Studies of PCa

Kesch et al. [11] aimed to evaluate FAP expression in tissue microarrays (TMAs, quantified through the H-index) from radical prostatectomy (RPE) or transurethral prostate resection (TURP) specimens of prostatic tissue from patients at different stages. They reported mean H-indexes for benign tissue of 0.018, for primary PCa of 0.031, for neoadjuvant androgen deprivation therapy (ADT) before RPE of 0.042, for CRPCa of 0.076 and for NEPCa of 0.051, indicating a significant rise in FAP expression as the disease progressed, as already described in a preclinical model [16]. They also imaged two patients with progressive mCRPCa and one with NEPCa and found [^68^Ga]Ga-DOTA-FAPi-04 describing multiple metastatic sites with high uptake, which confirmed their array’s results.

In a basket study, Giesel et al. assessed the biodistribution and preliminary dosimetry of [^68^Ga]Ga-DOTA-FAPi-02 and [^68^Ga]Ga-DOTA-FAPi-04 in 50 patients with various cancers including four cases of PCa using FDG as the standard of reference. A favourable biodistribution was found for both tracers, with fast tracer kinetics (prolonged retention for [^68^Ga]Ga-DOTA-FAPi-04). Both tracers were unaffected by the blood sugar levels and had similar dosimetry to other 68Ga compounds and [^18^F]FDG, enabling potential theranostics applications owing to the DOTA chelator. No specific data were reported for the patients with PCa [12].

A recent study by Greifenstein et al. [13] assessed the “kit-type” synthesis of the CAF-targeting tracer [^68^Ga]Ga-DATA5m.SA.FAPi and analysed the tracer’s uptake and biodistribution in one patient with sarcoidosis and five patients with various histologically confirmed metastatic malignancies, including one patient with PCa. With a very low physiological uptake in the brain, liver, intestine, bone and lungs, [^68^Ga]Ga-DATA5m.SA.FAPi showed a biodistribution similar to that of FAPi-02, but with less kidney uptake and blood pool activity and a higher uptake of [^68^Ga]Ga-DATA5m.SA.FAPi than FAPI-46 and FAPi-74 in the malignant lesions. The patient with PCa had metastatic small cell extensive disease (ISUP 3) and the highest [^68^Ga]Ga-DATA5m.SA.FAPi maximum standardised uptake value (SUV_max_ 13.7) in bone metastasis among the whole cohort.

Kratochwil et al. performed a retrospective analysis [19] to quantify the uptake of [^68^Ga]Ga-DOTA-FAPi-04 in 80 patients with 28 different tumour entities (confirmed by histopathological or unequivocal radiologic findings), including 4 PSMA-negative PCa cases. The authors defined low, intermediate and high uptake using SUV_max_ cut-offs of <6, from 6 to 12, and >12, respectively. In one PSMA-negative PCa case, the authors observed a high-intermediate [^68^Ga]Ga-DOTA-FAPi-04 uptake; however, in three NEPCa cases, the authors reported low uptake (SUV_max_ < 6), which is different from the findings of Vlachostergios et al., who reported high mRNA FAP expression levels in 21 patients with NEPCa who received previous treatment with taxanes and abiraterone or enzalutamide [17]. The main characteristics and results of the studies described in this paragraph are reported in Table 1.

### 4.2. Comparison of FAPi and PSMA PET for PCa

The aim of the study by Kessel et al. [18] was to compare FAP and PSMA expression levels in primary PCa by performing histological analyses and PET imaging in a small cohort of 11 PCa patients with Gleason scores of >7 (5 staging and 6 mCRPCa) and 3 patients with benign prostatic hyperplasia. Among the patients, the 6 patients with mCRPCa underwent both [^18^F]PSMA-1007 and [^68^Ga]Ga-DOTA-FAPi-46 PET/computed tomography (CT). The authors observed that the IHC FAPi expression level in the patients with prostatic disease was not necessarily found only in those with CAF, but also in those with benign myofibroblasts in the context of suspected chronic prostatitis. Nevertheless, The technique of [^68^Ga]Ga-DOTA-FAPi-46 PET/CT was useful for an advanced disease, as confirmed in a patient with NEPCa who had more FAPi-positive lesions than they did PSMA-positive lesions. This preliminary finding hampers the diagnostic relevance of FAPi in early PCa, as it is unspecific especially in the setting of prostatic inflammation.

In a case report, Pang et al. [27] described [^68^Ga]Ga-PSMA, [^18^F]FDG and [^68^Ga]Ga-DOTA-FAPi-04 PET/CT findings in a 65-year-old man with de novo mPCa and a PSA level > 60 ng/mL. [^68^Ga]Ga-PSMA PET/CT revealed osteogenic bony destruction with intense uptake throughout the skeleton, which suggested widespread metastatic bone lesions in the absence of any abnormal [^68^Ga]Ga-PSMA uptake in the prostate gland. With a time interval of 3 days, the patient underwent both [^18^F]FDG with only minimal prostatic uptake and [^68^Ga]Ga-DOTA-FAPi-04 PET/CT examinations, which detected non-PSMA/FDG-slightly avid primary PCa, which was later confirmed by a biopsy (Gleason score, 4 + 5, ISUP 5). The main characteristics and results of the studies described in this paragraph are reported in Table 2.

### 4.3. Comparison of FAPi and FDG PET for PCa

In an international retrospective multicentre analysis, Giesel et al. [20] compared different [^68^Ga]Ga-FAPi ligands with [^18^F]FDG PET/CT in terms of organ biodistribution and tumour uptake in various cancers (*n* = 71), including in one patient with PCa. The patients underwent both techniques within a median time interval of 10 days. In the absence of specific data for PCa, the authors reported that the [^68^Ga]Ga-FAPi uptakes in primary tumours and metastases were comparable with the [^18^F]FDG uptakes in most cases, but the SUV_max_ was significantly lower for [^68^Ga]Ga-FAPi than it was for [^18^F]FDG in the background tissues.

In a recent study, Lan et al. [21] compared the detection rate of [^68^Ga]Ga-DOTA-FAPi-04 with that of [^18^F]FDG PET/CT performed simultaneously in patients with various oncological (*n* = 110) and non-oncological lesions (*n* = 23), with histopathological or imaging/laboratory follow-ups being the gold standard. In the only patient with PCa (3.7 cm^3^) among the whole cohort, the authors reported lower SUV_max_ values for [^68^Ga]Ga-DOTA-FAPi-04 than they did for [^18^F]FDG PET/CT in the absence of other information (i.e., ISUP, therapies, disease stage and PSA levels). By contrast, for the only patient with prostatitis, the authors reported higher SUV_max_ values for [^68^Ga]Ga-DOTA-FAPi-04 than they did for [^18^F]FDG PET/CT in the absence of other characteristics (i.e., acute/chronic). However, in the whole cohort, except for those with myeloma and lymphoma, [^68^Ga]Ga-DOTA-FAPi-04 PET/CT showed superior detection efficacy for various primary and metastatic lesions and inflammatory diseases compared with those of [^18^F]FDG PET/CT.

A retrospective analysis by Wu et al. [22] compared the relative diagnostic efficacy of [^68^Ga]Ga-DOTA-FAPi-04 with that of [^18^F]FDG PET/CT in 30 patients with different cancers (3/30 patients with PCa) in terms of bone metastasis detection using imaging and clinical follow-ups as the reference standards. The authors observed that [^68^Ga]Ga-DOTA-FAPi-04 accumulation was significantly greater in both osteolytic and osteoblastic metastases, resulting in higher bone metastasis sensitivity than that of [^18^F]FDG PET/CT. In addition, [^68^Ga]Ga-DOTA-FAPi-04 PET/CT detected all 23 primary lesions, whereas [^18^F]FDG PET/CT did not visualise one primary renal cancer. However, [^68^Ga]Ga-DOTA-FAPi-04 detected 10 false-positive lesions, whereas [^18^F]FDG PET/CT visualised only 5 false positives.

In a case report, Xu T et al. [28] described the findings of [^68^Ga]Ga-DOTA-FAPi-04 and [^18^F]FDG PET/CT in a 76-year-old man with a history of chronic prostatitis, left shoulder osteoarthritis and a magnetic resonance imaging (MRI)-detected nodule in the left frontotemporal lobe, which was suspicious for brain metastasis. With a moderately elevated PSA level of 4.6 ng/mL, the authors observed abnormal uptake with both tracers in the known intracranial lesion, in the left shoulder and in the left peripheral zone of the prostate. A biopsy of the left posterolateral prostate region confirmed the presence of an ISUP 2 PCa. The patient also showed more tracer uptake at the site of the shoulder arthritis than that at the tumour site, indicating the potential of usefulness of [^68^Ga]Ga-DOTA-FAPi-04 imaging in cases with inflammation. In this case report, the increased [^68^Ga]Ga-DOTA-FAPi-04 uptake was similar to that shown by [^18^F]FDG imaging, which suggests that [^68^Ga]Ga-DOTA-FAPi-04 imaging may not be more tumour-specific than [^18^F]FDG is for PCa. The main characteristics and results of the studies described in this paragraph are reported in Table 3.

### 4.4. FAPi Theranostics Applications for PCa

In a case report, Khreish et al. [25] assessed a 77-year-old patient at the progression stage after two cycles of [^177^Lu]Lu-PSMA RLT. The restaging [^68^Ga]Ga-PSMA-11 PET/CT images demonstrated heterogeneous, modest uptakes of lymph nodes and bone metastases, whereas the [^18^F]FDG PET/CT image showed multiple additional lesions, indicating a clear FDG-positive/PSMA-negative mismatch pattern. Therefore, [^177^Lu]Lu-PSMA-617 was discontinued, and [^68^Ga]Ga-FAPi-04 PET/CT was performed to evaluate the potential for FAP-targeted therapy. The [^68^Ga]Ga-FAPi-04 PET/CT image depicted very intense tracer uptakes of all metastases (higher than those of PSMA and FDG), which suggests that in PCa, FAP-targeted radionuclide therapy (using molecules with longer retention times than those of FAPi-04) has the potential to overcome limitations related to tumour heterogeneity and an insufficient PSMA expression level.

In a small comparative study, Isik et al. [29] assessed two patients with mCRPCa who underwent [^68^Ga]Ga-PSMA, [^18^F]FDG and [^68^Ga]Ga-FAPi-04 PET/CT for the assessment of targeted RLT eligibility. The first patient’s PSMA PET/CT image showed that only that a part of the metastatic burden was detected with [^18^F]FDG and [^68^Ga]Ga-FAPi-04 PET/CT. In the second patient, [^68^Ga]Ga-FAPi-04 PET/CT outperformed [^18^F]FDG PET/CT in terms of the number of lesions detected and SUV values, but most of the lesions detected by PSMA PET/CT were FAPi-negative ones.

In a case report by Aryana et al. [30], a 70-year-old patient with mCRPCa underwent [^68^Ga]Ga-PSMA PET/CT for the evaluation of the presence of sufficient PSMA expression level for RLT, with [^177^Lu]Lu-PSMA enrolment. Numerous [^68^Ga]Ga-PSMA-avid skeletal metastases with low SUVs were found. However, owing to the low PSMA expression levels of the lesions, [^68^Ga]Ga-FAPi-46 PET/CT was performed for the evaluation of eligibility for FAPi-based RLT. The authors observed discordant findings between the [^68^Ga]Ga-PSMA and [^68^Ga]Ga-FAPi PET/CT scans regarding the detectability of lesions and SUVs, which suggests a potential role of FAPi RLT in patients with mCRPCa with insignificant PSMA expression levels or who underwent failed [^177^Lu]Lu-PSMA therapy. The work by Assadi et al. [23] aimed to assess the feasibility, safety and dosimetry of [^177^Lu]Lu-FAPi-46 RLT in a highly heterogeneous group of patients with different inoperable metastatic malignancies confirmed by histopathological examination. Twenty-one patients refractory to conventional therapies who had sufficient FAP expression levels in most lesions based on FAPi-based imaging (5 out of 21 patients underwent [^68^Ga]Ga-DOTA-FAPi-46 PET/CT, and 15 of 21 patients underwent [^177^Lu]Lu-DOTA-FAPi-46 pre-treatment scintigraphy) were enrolled. The treatment included increasing doses of [^177^Lu]Lu-DOTA-FAPi-46 (1.85–4.44 GBq) per cycle, with intervals from 4 to 6 weeks (median number of RLT cycles, 2; median injected activity in each cycle, 3.7 GBq). The RECIST criteria were uses for the response assessment. One of the nineteen finally included patients had PCa, and at the time of the analysis, they had been treated with a cycle of [^177^Lu]Lu-DOTA-FAPi-46 (1.85 GBq), with stable disease after 4.5 months of treatment. The dosimetric analysis revealed a minimal uptake rate in normal tissues and organs and an acceptable uptake rate with long retention in the tumoral regions (unlike the low retention time in FAPi-02 and FAPi-04). Therefore, the authors concluded that the findings of this preliminary investigation indicate the potential feasibility and safety of [^177^Lu]Lu-DOTA-FAPi-46 RLT for different aggressive tumours.

Finally, in the work by Fendler et al. [24], up to four cycles of [^90^Y]DOTA-FAPi-46 RLT were administered to patients with progressive metastatic malignancy with adequate haematopoiesis and high FAP expression levels (defined as [^68^Ga]Ga-DOTA-FAPi-46 SUV_max_ ≥ 10 in >50% of the tumour). Twenty-one patients, including 1 patient with mCRPCa, were found to be eligible for the treatment and received 47 [^90^Y]FAPi-DOTA-46-RLT cycles. The end point included RECIST and PERCIST responses after RLT, the overall survival (OS) and the FAP-RLT safety assessment. In the absence of specific PCa data, the authors concluded that despite the short retention time, [^90^Y]DOTA-FAPi-46-RLT was safe and led to stable disease in approximately one-third of patients, with prolonged survival. The main characteristics and results of the studies described in this paragraph are reported in Table 4.

## 5. Conclusions

Representing a wide spectrum of probes, [^68^Ga]Ga-DOTA-FAPi ligands have certain advantages for cancer assessment, including a high TBR, rapid renal clearance, fast tracer kinetics, no blood glucose level dependency and no need for resting.

It is well known that PCa is a highly heterogeneous disease, in which molecular imaging endorses a fundamental role. However, the characteristics of patients with low-PSMA-expressing PCa who may benefit from new probes are a “grey area” [31]. State-of-the-art literature regarding FAPi ligands in PCa is scarce, and this warrants further investigation, as only a few studies have assessed the clinical role of FAPi PET or FAPi RLT in the PCa scenario, with each cohort including only a maximum of 6 patients (for a total of 32 patients), providing scarce specific data [16,17,18,21]. In PCa theranostics, particular attention must be paid to the selection/use of FAPi ligands in terms of the retention time. Among the FAPi molecules that have been explored so far in PCa, FAPi-46 showed a longer retention time than FAPi-04 did, making it a more suitable ligand for RLT. The authors showed the feasibility of FAPi theranostics using a beta probe ([^90^Y]DOTA-FAPi-46) [24] and a probe with a longer half-life ([^177^Lu]Lu-FAPi-46) [23].

On the basis of our preliminary systematic review, we summarise that FAPi PET may potentially improve the diagnostic accuracy of next-generation imaging in patients with PCa with heterogeneous PSMA expression levels or at later stages of the disease. The role of FAPi PET for NEPCa is still controversial. Many issues remain to be solved, including enlarging the sample sizes of clinical studies, designing prospective studies with accurate methodologies to determine the exact diagnostic performance of FAPi PET in all phases of the disease and identifying ligands with a longer retention time in tumour tissues.

## Figures and Tables

**Figure 1 diagnostics-13-01175-f001:**
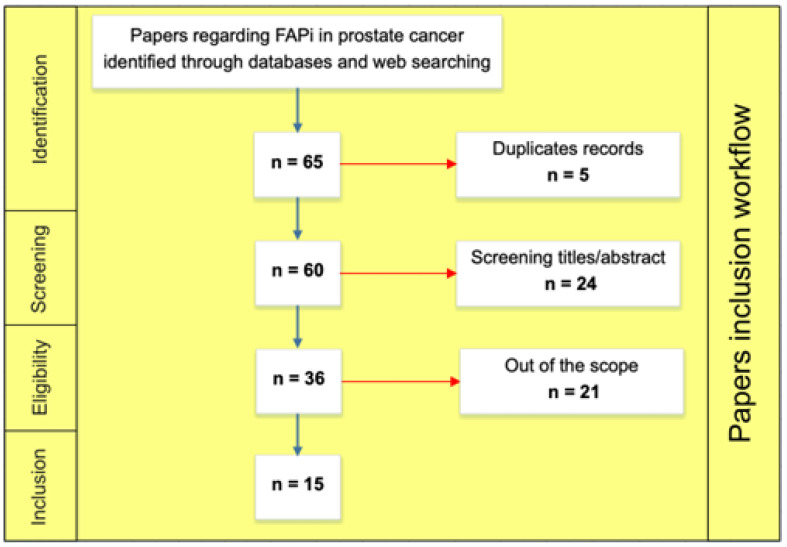
Paper inclusion workflow.

**Table 1 diagnostics-13-01175-t001:** Résumé of the described studies on FAPi IHC biodistribution for PCa.

Authors [ref.]	Year	FAPi Tracer	No. of Patients (PCa)	Scenario	ISUP	PSA (ng/mL)	Main Findings
Kesch et al. [11]	2021	[^68^Ga]Ga-DOTA-FAPi-04	94 (94 *)	PCa, mCRPCa, NEPCa	5 ***	NA	In the microarray analysis, the authors observed significantly higher H-index values for more advanced PCa (mCPRCa and NEPCa) than those of benign tissues and early PCa, as confirmed by imaging of [^68^Ga]Ga-DOTA-FAPi-04.
Greifenstein et al. [13]	2022	[^68^Ga]Ga-DATA^5m^.SA.FAPi	6 (1)	mPCa	3	NA	[^68^Ga]Ga-DATA^5m^.SA.FAPi had excellent diagnostic properties, detecting soft tissue and bone metastases, with high TBR and a remarkably high tumour SUV_max_ in mPCa (ISUP 3) with extensive disease.
Kratochwil et al. [19]	2019	[^68^Ga]Ga-DOTA-FAPi-04	80 (4 **)	PSMA-negative PCa	NA	NA	Intermediate-to-high [^68^Ga]Ga-DOTA-FAPi-04 uptake (SUV_max_ 6-12) was observed in a patient with PSMA-negative PCa. Low uptake (SUV_max_ < 6) was observed in 3 patients with NEPCa.

**Legend:** FAPi, Fibroblast activation protein inhibitor; FDG, fluorodeoxyglucose; ISUP, International Society for Urological Pathology; mCRPCa, metastatic castration-resistant prostate cancer; NA, not available; NEPCa, neuroendocrine PCa; PSMA, prostate-specific membrane antigen; SUV, standardised uptake value; TBR, tumour-to-background ratio; * Only 3 out of 94 patients underwent imaging with FAPi PET; ** 3 patients with NEPCa; *** Data available for only 1 patient.

**Table 2 diagnostics-13-01175-t002:** Résumé of the described studies that compared FAPi and PSMA PET for PCa.

Authors [ref.]	Year	FAPi Tracer	No. of Patients (PCa)	Scenario	ISUP	PSA (ng/mL)	Main Findings
Kessel et al. [18]	2021	[^68^Ga]Ga-DOTA-FAPi-46	14 (11 *)	PCa diagnosis and mCRPCa	5 (3) NA (3) **	23.8 [2.4–106] **	The role of FAPi in PCa diagnosis remains subject to further investigations, as it is also overexpressed in inflammatory diseases. Measurement of FAPi expression levels might be recommended as a complementary diagnostic tool for later PCa stages together with the use of PSMA and/or FDG or DOTATATE-PET.
Pang et al. [27]	2022	[^68^Ga]Ga-DOTA-FAPi-04	1 (1)	mPCa diagnosis	5	>60	The technique of [^68^Ga]Ga-DOTA-FAPi-04 PET/CT may be a useful imaging modality for the detection and localisation of non-PSMA avid primary PCa.

**Legend:** FAPi, Fibroblast activation protein inhibitor; FDG, fluorodeoxyglucose; mCRPCa, metastatic castration-resistant prostate cancer; PET/CT, positron emission tomography/computed tomography; PSMA, prostate-specific membrane antigen; SUV, standardised uptake value; * 6 out of 11 patients underwent FAPi PET; ** PSA median value with interquartile ranges for the 6 patients who underwent FAPi PET.

**Table 3 diagnostics-13-01175-t003:** Résumé of the described studies that compared FAPi and FDG PET for PCa.

Authors [ref.]	Year	FAPi Tracer	No. of Patients (PCa)	Scenario	ISUP	PSA (ng/mL)	Main Findings
Lan et al. [21]	2021	[^68^Ga]Ga-DOTA-FAPi-04	123 (1)	NA	NA	NA	In 1 patient with PCa, the authors observed lower SUV_max_ values for [^68^Ga]Ga-DOTA-FAPi-04 than those for [^18^F]FDG PET/CT.
Xu et al. [28]	2020	[^68^Ga]Ga-DOTA-FAPi-04	1 (1)	PCa diagnosis	2	4.6	In a patient with PCa (cT1c, ISUP 2), the [^68^Ga]Ga-DOTA-FAPi-04 PET/CT findings were similar to the [^18^F]FDG PET/CT findings, which indicates that [^68^Ga]Ga-DOTA-FAPi-04 may not be specific to PCa.

**Legend:** FAPi, Fibroblast activation protein inhibitor; FDG, fluorodeoxyglucose; ISUP, International Society for Urological Pathology; mPCa, metastatic prostate cancer; NA, not available; PET/CT, positron emission tomography/computed tomography; SUV, standardised uptake value; TBR, tumour-to-background ratio.

**Table 4 diagnostics-13-01175-t004:** Résumé of the described studies on FAPi applications in PCa theranostics.

Authors [ref.]	Year	FAPi Tracer	No. of Patients (PCa)	Scenario	ISUP	PSA (ng/mL)	Main Findings
Assadi et al. [23]	2021	[^68^Ga]Ga-DOTA-FAPi-46 and [^177^Lu]Lu-DOTA-FAPi-46	21 (2 *)	mPCa	NA	NA	RLT with [^177^Lu]Lu-DOTA-FAPi-46 is feasible, with dosimetry and toxicity values that are similar to those of [^177^Lu]Lu-DOTATATE and [^177^Lu]Lu-PSMA and an acceptable tumour retention time (up to 10 days after administration). In 1 patients with mPCa, the authors reported SD after 1 RLT cycle (1.85 GBq).
Fendler et al. [24]	2022	[^68^Ga]Ga-DOTA-FAPi-46 and [^90^Y]FAPi-46	21 (1)	mCRPCa	NA	NA	In spite of the short retention time, [^90^Y]FAPi-46 RLT is safe, with organ radiation doses below the critical range, high response rates, and prolonged survival in patients with mCRPCa.
Khreish et al. [25]	2019	[^68^Ga]Ga-DOTA-FAPi-04	1 (1)	mCRPCa	NA	NA	The technique of [^68^Ga]Ga-DOTA-FAPi-04 PET/CT revealed a higher detection rate than that of [^68^Ga]Ga-PSMA PET/CT and [^18^F]FDG PET/CT, opening new RLT opportunities for FAPi molecules in patients with highly dedifferentiated PCa, overcoming the limitation of PSMA expression heterogeneity.
Isik et al. [29]	2022	[^68^Ga]Ga-DOTA-FAPi-04	2 (2)	mCRPCa	NA	NA	FAPi molecules are promising novel tracers with theranostics applications in patients with mCRPCa, particularly those with heterogeneous tumour phenotypes.
Aryana et al. [30]	2022	[^68^Ga]Ga-DOTA-FAPi-46	1 (1)	mCRPCa	NA	1603	[^68^Ga]Ga-FAPi-46 theranostics may have a potential application in the treatment of patients with mCRPCa who have negative or low PSMA expression levels or failed [^177^Lu]Lu-PSMA therapy.

**Legend:** FAPi, Fibroblast activation protein inhibitor; FDG, fluorodeoxyglucose; IMRT, intensity-modulated radiation therapy; mCRPCa, metastatic castration-resistant prostate cancer; PET/CT, positron emission tomography/computed tomography; PSMA, prostate-specific membrane antigen; RLT, radioligand therapy; SD, stable disease; SUV, standardised uptake value; * One of the two patients did not undergo RLT.

## Data Availability

The data analysed during this review study are available from the corresponding author upon reasonable request.

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
