# Peer review of "Preliminary Findings of the Role of FAPi in Prostate Cancer Theranostics"

_diagnostics, 2023, doi:10.3390/diagnostics13061175_

Round 1

Reviewer 1 Report

The paper is useful to suggest new biomarkers in the field of prostate cancer diagnostics, since the reference biomarker(PSA) has a debatable diagnostic specificity and currently several works have been published to optimize the threshold of PSA testing for the prediction of PCa of advanced grade. Notably, current guideline report the need to improve the use of diagnostic tests for detecting advanced cancer.

This premise is relevant to increase the value of this research. Here I suggest some recent references:  

1.Individual risk prediction of high grade prostate cancer based on the combination between total prostate-specific antigen (PSA) and free to total PSA ratio.Clin Chem Lab Med. 2023 Jan 27.

2. Definition of Outcome-Based Prostate-Specific Antigen (PSA) Thresholds for Advanced Prostate Cancer Risk Prediction. Cancers 2021, 13, 3381.

In the tables relevant information lacks. In the tables  PSA median(and percentiles) levels at baseline  should be added  for each study.

Furthermore in the table you should add the % of PCa ISUP>=3(advanced cancer). In the text there is a blending of Gleason and Isup grade, I suggest to add the corresponding ISUP grade next to Gleason. 

Author Response

We sincerely thank the 3 reviewers for their time dedicated to our paper. All the modifications in the manuscript are highlighted in yellow.

Reviewer 1  

The paper is useful to suggest new biomarkers in the field of prostate cancer diagnostics, since the reference biomarker (PSA) has a debatable diagnostic specificity and currently several works have been published to optimize the threshold of PSA testing for the prediction of PCa of advanced grade. Notably, current guideline report the need to improve the use of diagnostic tests for detecting advanced cancer. This premise is relevant to increase the value of this research. Here I suggest some recent references: 1.Individual risk prediction of high grade prostate cancer based on the combination between total prostate-specific antigen (PSA) and free to total PSA ratio.Clin Chem Lab Med. 2023 Jan 27. 2. Definition of Outcome-Based Prostate-Specific Antigen (PSA) Thresholds for Advanced Prostate Cancer Risk Prediction. Cancers 2021, 13, 3381.  

R1Q1: In the tables relevant information lacks. In the tables  PSA median(and percentiles) levels at baseline  should be added  for each study. 

R1Q2: Furthermore in the table you should add the % of PCa ISUP>=3(advanced cancer). In the text there is a blending of Gleason and Isup grade, I suggest to add the corresponding ISUP grade next to Gleason.  

R1A1-2: We thank the reviewer for her/his kind, constructive and encouraging comments. We agree with the  issues and we now added ISUP and median PSA values (with IR ranges) in each table when available. In the text, we also added ISUP to GS in the period with gleason score.

Reviewer 2 Report

Nicely written paper. Reads well. 

The review did a research on manuscripts that deal with FAP or FAPi in prostate cancer. 15 papers were selected. These were studies with only a very low number of patients (i.e., 1-5 patients). Some of these were case reports. 

Definite conclusions can hardly be made on these data and on the comparison to other PET tracers. 

The authors should acknowledge that PSMA non-expressing tumors might not be different from those that express PSMA, especially in the hormone sensitive state [1,2]. So probably, PSMA non-expressing tumors do only exist in the castrate resistent state. 

Indeed as the authors state, the predictive value of FAPi can only be assessed and studied in larger patient samples and potentially in different disease stages. 

[1, 2] doi: 10.1111/bju.15896 doi: 10.1111/bju.15664. 

Author Response

We sincerely thank the 3 reviewers for their time dedicated to our paper. All the modifications in the manuscript are highlighted in yellow.

Reviewer 2 

Nicely written paper. Reads well.  

The review did a research on manuscripts that deal with FAP or FAPi in prostate cancer. 15 papers were selected. These were studies with only a very low number of patients (i.e., 1-5 patients). Some of these were case reports. Definite conclusions can hardly be made on these data and on the comparison to other PET tracers. 

R2Q1: The authors should acknowledge that PSMA non-expressing tumors might not be different from those that express PSMA, especially in the hormone sensitive state [1,2]. So probably, PSMA non-expressing tumors do only exist in the castrate resistent state. Indeed as the authors state, the predictive value of FAPi can only be assessed and studied in larger patient samples and potentially in different disease stages. [1, 2] doi: 10.1111/bju.15896 doi: 10.1111/bju.15664.  

R2A1: We thank the reviewer for her/his kind, constructive and encouraging comments. Indeed, the biggest concerns currently are focusing on PSMA dedifferentiated, late-stage PCa (i.e., neuroendocrine). It was, however, well documented by several authors that significant, PSMA-negative disease exists also in primary PCa (Dong-Xu Qiu et al. EJNMMI (2022) 49:2821–283, your CR); furthermore, the heterogeneity of primary PCa should not be underestimated and might lead to substantial underestimation of tumor extent within the prostate (Rüschoff et al. Eur J Nucl Med Mol Imaging. 2021 Nov;48(12):4042-4053). Therefore, the impression that PSMA-negative disease only exists in CRPCa might be biased. 

Reviewer 3 Report

The use of PET-CT in PCa assessment is increasing, with significant numbers of patients undergoing 68Ga-PSMA-11 PET-CT and more recently 18F-PSMA-DCFPyL PET-CT in Europe and Australia, while the frequency of such use is just picking up in the US. The number of cases of FAPi PET-CT in patients with PCa are few. The review the authors present in this manuscript is not focused, as per the title,  on PCa and the use of FAPi in a theranostic setting. Virtually all studies cited in this review are single case reports or larger studies that involve 1-4 patients with PCa.  The author's make statements that are clearly exaggerated and thus misleading.  For this paper to warrant publication, the integrity of reporting must be addressed and the paper's contents edited.  

There is significant redundancy in the text and the associated tables. 

There is a need for far better grammar throughout the paper.  Much of the sentence structure is awkward and some "sentences" are not sentences at all. I would consider this report as preliminary observations rather than a review. Only if the authors address the limitations of this study with the associated editing would I consider this worthy of publication. 

Author Response

We sincerely thank the 3 reviewers for their time dedicated to our paper. All the modifications in the manuscript are highlighted in yellow.

Reviewer 3  

The use of PET-CT in PCa assessment is increasing, with significant numbers of patients undergoing 68Ga-PSMA-11 PET-CT and more recently 18F-PSMA-DCFPyL PET-CT in Europe and Australia, while the frequency of such use is just picking up in the US. The number of cases of FAPi PET-CT in patients with PCa are few. The review the authors present in this manuscript is not focused, as per the title, on PCa and the use of FAPi in a theranostic setting. Virtually all studies cited in this review are single case reports or larger studies that involve 1-4 patients with PCa.  

R3Q1:The author's make statements that are clearly exaggerated and thus misleading.  

For this paper to warrant publication, the integrity of reporting must be addressed and the paper's contents edited. There is significant redundancy in the text and the associated tables.  

There is a need for far better grammar throughout the paper.  Much of the sentence structure is awkward and some "sentences" are not sentences at all. 

I would consider this report as preliminary observations rather than a review. Only if the authors address the limitations of this study with the associated editing would I consider this worthy of publication.  

R3A3: Being a new and potentially game-changing probe, and despite the paucity of data in the literature, we tried to present a complete assessment of the FAPi PET applications in PCa so far, also including theranostic applications. However, we agree with the reviewer that there is redundancy in the text and tables that we tried to reduce, also improving grammar. Finally, we also modified the title and the conclusion section accordingly.  

Round 2

Reviewer 1 Report

In my previous evaluation I had recommended to report about the evidence that current guideline report the need to improve the use of diagnostic tests as PSA for detecting advanced cancer. This premise is relevant to increase the value of this research!!!. In introduction and discussion this should be considered, since PSA is the first recommended diagnostic test and the evidence that in the single studies PSA levels have not beeen retrieved opens a discussion about the "good" quality of these studies.

Here I suggest some recent references: 1.Individual risk prediction of high grade prostate cancer based on the combination between total prostate-specific antigen (PSA) and free to total PSA ratio.Clin Chem Lab Med. 2023 Jan 27. 2. Definition of Outcome-Based Prostate-Specific Antigen (PSA) Thresholds for Advanced Prostate Cancer Risk Prediction. Cancers 2021, 13, 3381.  

Author Response

It is well-established that PSA is a non-invasive technique to assess PCa, from screening through diagnosis (with DRE, MRI, and biopsy) to follow-up. However, in a diagnostic journal, and after a brief introduction, through our manuscript we tried to highlight the preliminary data about FAPi theranostics application in PCa which are, so far, mainly limited to advanced and metastatic PCa in which PSA values are “secondary”, also considering that optimal intervals for PSA follow-up are unknown [Mottet N et al. EAU-EANM-ESTRO-ESUR-SIOG Guidelines on Prostate Cancer-2020 Update. Part 1: Screening, Diagnosis, and Local Treatment with Curative Intent. Eur Urol. 2021 Feb;79(2):243-262. doi: 10.1016/j.eururo.2020.09.042. Epub 2020 Nov 7. PMID: 33172724]. This could also probably explain why PSA values are missing in almost each presented paper included in our preliminary review. We already have read the reviewer’s 1 first round suggestions also regarding the 2 references which were “indirectly” invited to be added to our manuscript. Indeed, we deeply assessed both papers and we did not find any relationship with our manuscript. Also, the handling Editor and Reviewers should take into account the “MDPI instruction for reviewers” policy (https://www.mdpi.com/reviewers) as follows: “Reviewers must not recommend excessive citation of their work (self-citations), another author’s work (honorary citations) or articles from the journal where the manuscript was submitted as a means of increasing the citations of the reviewer/authors/journal. You can provide references as needed, but they must clearly improve the quality of the manuscript under review.”

Reviewer 3 Report

I have spent hours on multiple days reviewing and extensively editing this article. Maybe other reviewers do not take a review for the peer-reviewed literature as seriously as they should.  I find this paper to be of little value given the paucity of cases where FAPi PET is compared to the current standard using PSMA PET-CT.  Case reports comparing FAPI PET-CT with FDG PET-CT are not of value given the essential global consensus that FDG PET-CT in PCa is seldom  used to assess PCa.  There is just not enough data here to warrant this paper being published. 

Moreover, the editors of MDPI journals must insist that authors who do not use English as their primary written language should seek advice from those with expertise in English grammar. So much of this paper is hard to understand due to poor grammar,especially awkward sentence structure, and often the wrong choice of words.  I have submitted my recommendations up to page 7 of the manuscript, but I have no patience to spend another 4 hours to try to evolve this basically empty article. Not only are there a limited number of cases of PCa reported here, but the comparison of FAPI PET with FDG does not warrant publishing this article.  

Author Response

We are aware that this paper deals with new applications of a new molecule. However, we hope that other readers may find this paper useful as the handling editors and the other 2 Reviewers. We are sorry about Reviewer 3 time spent editing our paper and we are grateful for the suggested form improvements. Here are the accepted/followed improvements:

The paper has now a new title as suggested

In the whole manuscript, we changed the word “theranostic” with “theranostics”

We added almost every modification in the manuscript as gently suggested.

Furthermore, we are very sorry that our modifications were not enough to improve our article; therefore, we sent the manuscript to a certified native English specialist “scribendi” who proofread and edited the paper.